# Effectiveness of Operation K9 Assistance Dogs on Suicidality in Australian Veterans with PTSD: A 12-Month Mixed-Methods Follow-Up Study

**DOI:** 10.3390/ijerph20043607

**Published:** 2023-02-17

**Authors:** Melissa Sherman, Amanda D. Hutchinson, Henry Bowen, Marie Iannos, Miranda Van Hooff

**Affiliations:** 1Justice and Society, University of South Australia, Adelaide 5072, Australia; 2Military and Emergency Services Health Australia (MESHA), The Hospital Research Foundation Group, Adelaide 5000, Australia; 3Allied Health and Human Performance, University of South Australia, Adelaide 5000, Australia; 4Adelaide Medical School, Faculty of Health and Medical Sciences, The University of Adelaide, Adelaide 5000, Australia

**Keywords:** PTSD, assistance dog, suicidality, depression, anxiety, veteran

## Abstract

Post-traumatic stress disorder (PTSD) is a pervasive disorder among both current and ex-serving Australian Defence Force (ADF) members. Studies have shown current psychological and pharmacological treatments for PTSD are suboptimal in veterans, with high dropout rates and poor adherence to treatment protocols. Therefore, evaluating complementary interventions, such as assistance dogs, is needed for veterans who may not receive the ultimate benefit from traditional therapies. The present longitudinal mixed-method study examined the effectiveness of Operation K9 assistance dogs among sixteen veterans with PTSD, specifically, their effects on suicidality, PTSD, depression, and anxiety from baseline to 12 months post-matching. Self-reported measures were completed prior to receiving their dog (baseline) and at three time points (3, 6, and 12 months) following matching. The Clinician-Administered PTSD Scale for DSM-5 was used to assess the severity of every PTSD case. Veterans participated in a semi-structured interview 3 months post-matching. Whilst there was a reduction in the proportion of veterans reporting any suicidality, there was no significant change in the probability of veterans reporting suicidality between time points. There was a significant effect of time on PTSD, depression, and anxiety symptoms. Three major themes emerged from qualitative data analysis: life changer, constant companion, and social engagement. Qualitative data suggest assistance dogs can have a positive impact on important areas of daily life and support veterans in achieving some of the prerequisites for health, including access to services, transport, education, employment, and development of new and diverse social and community connections. Connections were key in improving health and wellbeing. This study exemplifies the power of human–animal relationships and adds emphasis to the need to take these seriously and create supportive healthy environments for veterans with PTSD. Our findings could be used to inform public health policy and service delivery, in line with the Ottawa Charter action areas and indicate that for veterans with PTSD, assistance dogs may be a feasible adjunct intervention.

## 1. Introduction

Post-traumatic stress disorder (PTSD) is a pervasive disorder among both current and ex-serving Australian Defence Force (ADF) members, resulting in many disabling negative outcomes [1]. Approximately 5000 ADF members transition from Regular service each year, either transferring to active or inactive Reserves or discharging completely [2]. It is estimated that 46% of recently transitioned ADF members meet diagnostic criteria for a mental disorder, with an estimated 17.7% and 24.9% meeting criteria for 12-month and lifetime PTSD, respectively [2].

PTSD has been associated with increased rates of morbidity, mortality, and suicidality (suicidal thoughts, plans, and attempts) and is commonly comorbid with depression and anxiety in the general population [3,4,5]. In military populations, the 2018 Transition and Wellbeing Research Programme found 80% of recently transitioned ADF members who met the 12-month criteria for PTSD had another comorbid mental disorder [2]. Similarly, a study of US veterans found those who experienced suicidal ideation had a greater likelihood of screening positive for PTSD and depression [6].

Between 2001 and 2018, there were 267 certified suicide deaths among ADF ex-serving personnel [7]. When compared to the age-adjusted Australian population, the rate of completed suicide is 21% higher for ex-serving males and 127% higher for ex-serving females [7]. In the Transition and Wellbeing Research Programme, over 20% of transitioned ADF members reported suicidal ideation, plans, or attempts in the past 12 months [2]. Given the evidence suggesting PTSD presents a risk factor for suicidality, identifying ways in which this can be reduced is critical to preventing suicide deaths [8]. 

Individuals with PTSD who do not engage in treatment are at risk of a chronic course of illness [9]. Treatments with the strongest evidence for PTSD include eye movement desensitisation and reprocessing (EMDR), cognitive processing therapy (CPT), and prolonged exposure (PE) [10]. However, many veterans who have undergone these psychotherapies still meet the diagnostic criteria for PTSD and continue to experience symptoms [11,12,13]. A review examining the effectiveness of CPT and PE in defence and veteran populations found that two-thirds maintained a PTSD diagnosis after treatment [13]. Other studies have reported non-response and dropout rates of up to 50% for veterans [14,15].

As such, there is a clear need to evaluate the use of adjunct interventions as a supplement to traditional therapies. An emerging area is the use of assistance dogs as an adjunct intervention for veterans with PTSD. An assistance dog is specifically trained to perform a variety of tasks aimed at alleviating PTSD symptoms [16,17]. Tasks include interrupting episodes of stress and anxiety and supporting independence and social interaction [18].

A systematic review of the effect of assistance dogs on PTSD symptoms in veterans concluded that participants experienced improved wellness because the dog had a calming influence on their mental/emotional state in potentially stressful situations [19]. Other studies report that assistance dogs may minimise the symptoms of PTSD by providing a sense of safety and confidence, increasing community engagement, and supporting independence [12,17,20]. A small number of studies have evaluated the impact of assistance dogs on veteran wellbeing, reporting reductions in PTSD, depression, and anxiety symptoms and improved quality of life and social functioning 3 to 6 months after matching a dog with a veteran [17,21,22,23]. Kloep et al., (2017) demonstrated a clinically significant change in the PTSD Checklist (PCL) symptom severity score and a reliable improvement in PTSD symptoms in 91.7% of (*n* = 12) veterans 6 months after they received their assistance dog [21].

Qualitative studies frequently reported reductions in suicidality, with veterans accrediting their assistance dog with keeping them alive [24,25,26,27]. McLaughlin and Hamilton (2019) explored the influence of service dogs on participation in daily occupations in Australian veterans with PTSD [26]. Veterans reported having an assistance dog decreased their suicidal ideation, and one individual stated their dog had prevented them from carrying out their suicide plan. However, these prior studies did not collect quantitative measures of mental health or follow veterans over time, thus preventing the assessment of clinical changes in wellbeing, and making it difficult to fully assess the impact of assistance dogs.

Despite limited empirical evidence, the demand for and use of assistance dogs is increasing in Australia. Therefore, it is critical to investigate the effectiveness of assistance dogs as an adjunct intervention for veterans with PTSD within the Australian context. Whilst preliminary evidence supports the use of assistance dogs as an adjunct intervention, methodological differences and the small number of studies available make it difficult to compare results. Most studies to date also tend to rely on standardised self-reported measures that do not examine the lived experience of the veteran or are purely qualitative. Furthermore, few studies have used a longitudinal approach to measuring changes over time. A longitudinal mixed-method study design incorporating a more objective diagnostic measure of PTSD, such as the Clinician-Administered PTSD Scale (CAPS-5) [17,28], would be valuable in providing a more holistic investigation of the effectiveness of assistance dogs in helping veterans with PTSD cope.

This study is the first in Australia to employ both quantitative and person-centred qualitative methods to investigate the effectiveness of an assistance dog program in helping Australian veterans with PTSD cope. The aim is to longitudinally examine the effectiveness of Operation K9 assistance dogs on mental health and wellbeing in veterans with PTSD over a 12-month period. This study will address the following research questions: 1. What is the effect of having an Operation K9 assistance dog on veterans’ mental health and wellbeing (specifically suicidality, PTSD, depression, and anxiety), as measured by both self-reporting and clinician-administered measures of mental health over a 12-month period? and 2. What are veterans’ perceptions and experiences of having an Operation K9 assistance dog in terms of mental health and wellbeing?

## 2. Materials and Methods

### 2.1. Participants

This study analysed data from 16 veterans who received an assistance dog through the Royal Society for the Blind Operation K9 Program between October 2015 and March 2019.

### 2.2. Operation K9 Program

The Operation K9 Program provides assistance dogs to Australian Defence Force (ADF) veterans (ex-serving) with PTSD. Individuals are signposted to the program either through a veteran’s support agency, medical and allied health professionals, or self-referral. A small breeding program supplies the Operation K9 Program. A considered matching process of client and dog ensures that working teams worked optimally and within their strengths. The rigorous application process takes many factors into consideration, including physical abilities, verbal capacity to interact with an assistance dog, and information from medical professionals, specialist skills of each dog, the needs of veterans, and the personalities of both dog and human. Whilst providing companionship, comfort, support, and increased social interaction, the Operation K9 assistance dogs are trained to meet the specific individual needs of each veteran. Given the process of breeding, training, and matching dogs is time- and resource-intensive, the total number of participants in this study was small due to constraints in the number of dogs available to be trained. Further information can be found on the Operation K9 website (https://www.seedifferently.org.au/operation-k9-steps-up (accessed on 6 February 2020).

### 2.3. Design

This was a mixed-method, within-group, repeated-measures study. The independent variable was time (baseline, 3, 6, and 12 months post-matching). The dependent variables were self-reported scores on suicidality, PCL-5 (PTSD Checklist), DASS-21 (Depression Anxiety and Stress Scale), depression and anxiety subscales, and clinician-assessed PTSD using the CAPS-5 (Clinician-Administered PTSD Scale for DSM-5). Figure 1 displays assessment timings.

### 2.4. Ethics Approval

The research was approved by the Department of Defence and Veterans’ Affairs Human Research Ethics Committee (ethics approval number DVAHREC #E015/005) and the University of South Australia’s Human Research Ethics Committee (ethics approval number 203074).

### 2.5. Materials and Measures

#### 2.5.1. Self-Report Questionnaire

At baseline, 3, 6, and 12 months, all participants completed a self-report questionnaire containing questions on demographic information and validated measures of mental health and wellbeing. Suicidal ideation plans and attempts were examined using four questions from the National Survey of Mental Health and Wellbeing [29]. Participants responded yes or no to four questions: (1) In the last 3 months, have you ever felt that life was not worth living? (2) In the last 3 months, have you ever felt so low that you thought about committing suicide? (3) In the last 3 months, have you made a suicide plan? (4) In the last 3 months, have you attempted suicide? To indicate whether participants reported any suicidality item or no suicidality items, a dichotomous variable was created. The proportion of respondents indicating any suicidality was analysed and reported.

The PTSD Checklist (PCL-5) was used to assess PTSD symptom severity [30]. The scale consists of 20 self-report items; participants are asked to rate on a 5-point Likert scale how much they experienced each symptom over the last month from 0 (not at all) to 4 (extremely) [31]. Scores for each item are summed to give a total symptom severity scale score between 0 and 80 [31]. Cronbach’s alpha for the current study was computed (α = 0.96).

Depression and anxiety symptom severity were measured using the Depression Anxiety Stress Scale—Short Form (DASS-21) depression and anxiety subscales [32]. The DASS-21 consists of 21 items (7 items per subscale); participants are asked to rate how much a statement (e.g., ‘I felt downhearted and blue’) applied to them over the past week on a 4-point Likert scale ranging from 0 (did not apply to me at all) to 3 (applied to me very much or most of the time) [32]. Scores range from 0 to 42; high scores indicate higher levels of depression and anxiety. In this study, Cronbach’s alpha for the depression and anxiety subscales was 0.93 and 0.89, respectively.

#### 2.5.2. Clinical Interview

A clinical diagnosis of PTSD was obtained at baseline, 6, and 12 months via a structured diagnostic interview using the Clinician-Administered PTSD Scale for DSM-5 Past Month Version (CAPS-5) [33]. PTSD diagnostic status is determined in line with DSM-5 diagnostic criteria after dichotomising individual symptoms as ‘present’ or ‘absent’ [33]. If a symptom score is rated as 2 (moderate) or higher, a symptom is considered present [33]. In the present study, the proportion of participants meeting the diagnostic criteria for PTSD was analysed and reported.

#### 2.5.3. Semi-Structured Qualitative Interview

Semi-structured qualitative interviews were undertaken three months post-matching. Open-ended interview questions enabled unrestricted responses, therefore providing richness and depth of data [34]. Participants were asked about the positives and challenges of having their dog, as well as changes in mental health and wellbeing. Specific questions about suicide were not asked; however, the topic emerged during the interviews.

#### 2.5.4. Procedure

Participants were provided with information regarding the study during their first meeting with Operation K9 staff and asked to provide consent to be contacted by the research team. A member of the research team contacted potential participants; consenting participants completed a participant consent form.

Participants were asked to complete the self-report questionnaire at four time points: (1) treatment baseline (about to receive dog), (2) 3 months, (3) 6 months, and (4) 12 months post-matching. Participants were provided a list of resources and contacts within the questionnaire in case they felt distressed by any aspect of the study. All interviews were conducted in person or via telephone and audio-recorded to allow transcription. Interviews also included opportunities for feedback and for any concerns to be raised. The study’s interview protocol also included detailed procedures to support and manage potentially distressed and suicidal participants.

#### 2.5.5. Data Analysis

##### Research Question 1

Statistical analyses were conducted using IBM SPSS Statistics (Version 26). Statistical significance was considered at *p* < 0.05. A sensitivity power analysis was performed using G*Power3 software [35] with α = 0.05, power = 0.80, and *n* = 16 to determine the minimum effect size the study was sensitive to. The study was sufficiently sensitive to detect a medium to large effect size. Continuous variables and residual errors were normally distributed with no outliers (according to histograms, skewness, and kurtosis values, and ZPRED and ZRESID values [36]).

Total scores for PCL-5 and DASS-21 depression and anxiety subscales were calculated using mean imputation to replace missing data (<20%). Following imputation, one baseline PCL-5 score was missing, and so mixed-effect models, a model well suited for dealing with missing data, were conducted with the missing data points [37]. For calculation and reporting of means and standard deviations, the missing score was interpolated. Data for one baseline and one 6-month CAPS-5 assessment were missing, and these were excluded from analysis.

Generalised estimating equation models were conducted to examine changes in the probability of suicidality and severity of PTSD case (CAPS-5) across time. The models specified a dichotomous variable of suicidality and PTSD case where 1 = Yes and 0 = No, with a predictor variable of time (baseline measurement reference category) and panel variable of subject ID.

To account for repeated measurements of self-reported PTSD symptoms (PCL-5) and depression and anxiety symptoms (DASS-21 subscales), scores were compared over time using linear mixed-effects models. The main effect of time is reported. Planned comparisons were conducted to investigate statistically significant changes in scores and Cohen’s *d* effect size between baseline and the other time points. Cut-offs for effect size were calculated using the Cohen’s *d* method with values of small (0.20), medium (0.50), and large (0.80).

Reliable change and clinical significance were evaluated for PTSD symptom severity (PCL-5) from baseline (BL) to 3 months, 6 months, and 12 months to estimate the extent to which symptom changes could be considered reliable and clinically significant. Evidence suggests reliable change is represented by a 5–10-point change and a clinically significant change by a 10–20-point change in PCL-5 total score [30]. The current study used the minimum thresholds of 5 and 10 points, respectively.

##### Research Question 2

Qualitative analysis was used in a complementary way to elaborate on the results from the quantitative analysis and increase understanding and meaningfulness [38]. To identify and interpret patterns in the qualitative data, the Braun and Clarke (2006) six-phase thematic analysis method was used [34]. Four participants were excluded from qualitative data analysis due to missing data. Interviews were transcribed verbatim and checked for accuracy. Initial codes and themes were generated by the researcher, discussed with the research team (to increase the trustworthiness of the qualitative analysis; [39]), and then refined, and findings were reported.

## 3. Results

Participant demographic data for sex, age, and service are presented in Table 1.

### 3.1. Self-Report Questionnaire

#### 3.1.1. Suicidality

The proportion of participants reporting suicidality at each time point is presented in Figure 2. Although there was a 34% reduction in the proportion of participants reporting suicidality from baseline to 3 months, the change in cases across all time points was not significant (Table 2).

#### 3.1.2. PTSD, Depression, and Anxiety

Means and standard deviations for the PCL-5 symptom severity score and DASS-21 depression and anxiety subscale scores are presented in Table 3.

Mixed-model analysis indicated a significant effect of time on self-reported PTSD symptom severity: *F*(3, 44.029) = 20.l29, *p* < 0.001. Planned comparisons revealed a significant mean difference between baseline and 3, 6, and 12 months with effect sizes ranging from 0.43 to 0.94 (Table 4). Results indicate that 12 months after receiving their assistance dog, 88% of participants achieved a reliable change in PTSD symptoms and 63% reached a clinically significant change (Table 5).

Mixed-model analysis indicated a significant effect of time on depression: *F*(3, 45) = 4.461, *p* = 0.008. Planned comparisons revealed a significant mean difference between baseline and 3, 6, and 12 months (Table 4). There was a small increase in the mean score between 6 and 12 months; however, this change was not significant.

Mixed-model analysis indicated a significant effect of time on anxiety: *F*(3, 45) = 11.642, *p* < 0.001). Planned comparisons revealed a significant mean difference between baseline and 3, 6, and 12 months with large effect sizes (Table 4).

### 3.2. Clinician-Diagnosed PTSD (CAPS-5)

At baseline, 10 (67%) participants met the diagnostic criteria for PTSD, while at 6 months there were 7 (47%) and at 12 months a total of 5 (31%). Overall, there was a 50% reduction in the number PTSD cases between baseline and 12 months with a significant change observed (Table 6).

### 3.3. Semi-Structured Qualitative Interview

Data from qualitative interviews were analysed to examine veterans’ perceptions and experiences of having an Operation K9 assistance dog. Three major themes emerged from the data analysis: life changer, constant companion, and social engagement. Themes, findings, and illustrative data examples are presented.

#### 3.3.1. Life Changer

This theme captured the life-changing impact of assistance dogs in improving quality of life for veterans and providing support for symptom management:


*‘See I didn’t even think it was possible for me to ever, ever improve. I thought this was going to be my lifestyle for the rest of my life, but you know this has just absolutely proven me wrong, it’s turned me right around 180 degrees’.*
(Veteran #12)

Three veterans reported the role of their dog in suicide prevention, with one stating *‘cause otherwise I wouldn’t be here’* (Veteran #6). Another veteran, talking about suicidality, said, *‘this has now alleviated all of that, I don’t even think about it anymore’* (Veteran #12). Furthermore, veterans expressed a sense of responsibility to care for their dog as providing a reason to live, for example:


*‘You just sort of look at (dog) and you think, well, you know, at least she’s worth living for and she’s there … it sort of reminds you constantly, you know, when you’re all by yourself, there’s always someone there.’*
(Veteran #6)

and *‘I need to stay alive now otherwise, (dog) would miss me … who would look after (dog), he gives me that right to live again’ (Veteran #12). For many veterans their assistance dog had given them a sense of purpose. One said their dog gave them ‘a reason to be’ (Veteran #2), while another ‘it gives me a reason and purpose now, so I’ve got to stay well and fit enough now to look after him, so he’s now my family’*(Veteran #12).

Veterans reported a sense of having reclaimed their life, giving them independence and a way of managing their emotional and mental states and coping with a range of symptoms including anxiety, low mood, and hypervigilance. One veteran said, *‘I notice I don’t do it as much (hypervigilance) and even my seating its, you know, I just go oh there’s some empty seats, we’ll go sit there, you know have my back exposed …’* (Veteran #14). While another said, *‘if I’m feeling a bit depressed or whatever he’s there, and I know he’s there and it just takes my mind off things makes life a lot easier’* (Veteran #9). Veterans reported their dog actively seeking them out to interrupt symptoms provided comfort, resulting in them feeling calm and relaxed. One described their dog as their *‘comfort blanket’* (Veteran #4), while another noted:


*‘He calms me and he knows, he really does know when you’re having a bad day or a problem... he just lays and puts his head on my foot and it is the most calming, relaxing feeling and it’s a confidence thing you know, it just makes me feel good all the time’.*
(Veteran #12)

#### 3.3.2. Constant Companion

This theme reflected upon the unique bond that is characterised by the human–dog relationship. Veterans highly valued the constant physical and emotional presence of their dog, appreciating how their dog was ‘always there’ for them. Assistance dogs were described as providing a stable base of support for veterans and providing a buffer against stressful and anxiety-producing situations.

A number of veterans also reported the presence of their dog provided a sense of safety and security, one stating they are *‘in safe hands’* (Veteran #12), while another referred to their dog as their *‘security blanket’* (Veteran #5). Veterans reported the sense of being looked after by their dog, for example: *‘… constant sort of watching on her part and constant support’* (Veteran #1).

The presence of their dog seemingly provided a calming and therapeutic effect. Many veterans discussed feeling increased self-confidence due to the presence of their dog, and how this has enhanced their communication skills, while others said their dog was a source of reassurance, for example, *‘giving me more reassurance and confidence … the ability to rest easier at night, the comforting of just walking around with me during the day … it is all been absolutely amazingly positive’* (Veteran #13).

#### 3.3.3. Social Engagement

Prior to receiving their assistance dog, many veterans reported difficulties with isolation and disengagement from family, friends, and community: *‘I was a recluse and didn’t leave my home for many, many years and now every day is an adventure … and gives me something to look forward to*’ (Veteran #12). Having their assistance dog resulted in positive interactions with family, friends, and members of the community, leading them to feel more comfortable in public and allowing them to spend increased time in the community and engage more socially. For example, one veteran noted *‘I’ve just felt a lot more comfortable going out in public … kind of broken that barrier where I used to … try and avoid the public at all costs.*’ (Veteran #9). Another stated that *‘I can like sit in a cafe and not feel self-conscious anymore … I won’t race off I’ll just linger and try and relax …’* (Veteran #1).

## 4. Discussion

The aim of this study was to longitudinally examine the effectiveness of Operation K9 assistance dogs on mental health and wellbeing in veterans with PTSD over a 12-month period. Although there was no significant change in suicidality, there was a significant reduction in the proportion of veterans meeting clinician-diagnosed PTSD and reduced severity of self-reported PTSD, depression, and anxiety. Veterans’ perceptions of their assistance dog’s role in supporting their mental health and wellbeing were classified into three main themes: life changer, constant companion, and social engagement.

This is the first longitudinal mixed-method study investigating an Australian assistance dog program for veterans with PTSD. Only one other study has investigated changes in suicidality of participants after receiving an assistance dog [23], with results consistent with ours. In both studies, although reductions were seen in the proportion of veterans reporting suicidality, these reductions were not significant. It is possible, however, that this was due to the small sample size and the low prevalence of suicidality in the study population. Hence, results may reach statistical significance with a larger sample.

A contribution and strength of the current study is the rich personal experiences provided by veterans about the impact of their assistance dog on their mental health and wellbeing. In relation to suicidality, veterans reported that their assistance dogs had given them a sense of purpose and responsibility, a ‘reason to be’, and a reason to live. Three veterans explicitly discussed the role of their dog in reducing suicidality. This is consistent with previous qualitative studies that have reported the role of an assistance dog as a protective factor for suicidality [25,26]. Interestingly, participants in our study did not raise concerns regarding experiencing challenges, such as managing their dog’s needs and feeling overwhelmed when in public spaces due to the attention they received when they went out with their dog, as was found in a previous study [27].

The current study is the first study to suggest that assistance dogs may provide a sustained reduction in symptoms of PTSD, depression, and anxiety over a 12-month period. Findings are consistent with previous quantitative studies that found a significant reduction in self-reported PTSD, depression, and anxiety symptoms in participants 3 to 6 months after receiving an assistance dog [17,21,22,23].

To account for potential self-report response bias for PTSD, the CAPS-5 was used in the current study to provide an objective, clinical measure of PTSD based on DSM-5 criteria. This is the first study to use a validated clinician assessment of PTSD in an evaluation of assistance dogs. Our results suggest that whilst veterans may still have some symptoms of PTSD, depression, and anxiety, they may experience reduced symptom severity and distress due to the presence of their assistance dog. The significant decreases seen in PTSD, depression, and anxiety symptoms may be contributing to increased functioning, independence, and social engagement. For example, veterans acknowledged that even though they may be feeling depressed or anxious, having their dog there ‘just makes it easier’. Veterans emphasised their relationship with their dog above all else, describing their dog as a constant, stable base of support, providing an overall calming effect. These findings are consistent with those of Van Houtert et al. (2018), which suggest that improvements in PTSD symptoms are due to the dogs’ calming influence on mental and emotional states in stressful situations [19]. Whilst not conclusive, these results suggest it is possible assistance dogs may, through interrupting and halting the progression of symptoms, increase veterans’ functioning.

### Limitations and Future Directions

A limitation of the current study was the lack of a control group of veterans with PTSD not receiving an assistance dog. Recruiting such control groups is, however, extremely difficult in longitudinal research, particularly when participants are required to complete assessments over multiple time points, with no benefits to them. Future studies could consider including a waitlist control group to provide a comparison group and enable an investigation of the efficacy of assistance dogs.

The veterans recruited for this study were considerably motivated to have an assistance dog to help manage their PTSD-related symptoms, and as a result may not be representative of all veterans with PTSD. As noted by O’Haire and Rodriguez (2018) in their paper, this limits generalisability to those who are amenable to receiving an assistance dog. Furthermore, the small sample size due to low availability of assistance dogs limits the generalisability of the findings from this study to other groups of veterans. While the current study was adequately powered for large effects, future research should aim to use a larger sample. The cost and time it takes to train an assistance dog may impact the extent to which studies could be implemented in the future.

The current study did not specifically ask veterans what challenges they experienced as a result of having an assistance dog. Future research should explore this in more detail and be used to inform policy and service delivery [27].

Given the high attrition rates in other interventions for veterans’ mental health [14,15], further investigation into the low attrition rate reported for this study (16%; three participants) is warranted. It is possible that assistance dogs may offer a solution to high attrition rates, as seen in other treatments. As the qualitative data were analysed only 3 months after veterans received their assistance dog, we were unable to investigate if their perspectives and experiences of having an assistance dog changed over a longer time period.

One concern is that the decreases seen in the current study are partly due to a ‘repeated measurement effect’. This has been observed in psychological research when measures of the self-reported negative effect decrease over time, even without treatment [40]. While some people may just ‘get better’ over time, evidence from a study by Monson et al. (2006) comparing a group of veterans treated with cognitive processing therapy to a control group found that PTSD symptoms in the control group remained stable across time. In our study, the sustained reduction in symptoms across time from baseline to 12 months suggests assistance dogs are an active component accounting for the changes reported and the effects seen may not be due to novelty or the attention the veteran received from the trainer when training their assistance dog [17,27,41].

The use of a dichotomised variable for suicidality was somewhat limiting in terms of statistical power. Questions about suicidality were not explicitly asked in the qualitative interviews but rather, related content emerged organically. This may mean that some veterans did not speak about the role of their assistance dog in terms of reducing suicidal thoughts and behaviour potentially due to reluctance to disclose such thoughts and behaviours. Future studies could include a more comprehensive continuous measure of suicidality to increase the statistical power to observe changes. It may also be of interest to include specific questions about suicidality in qualitative interviews to understand veterans’ perceptions and experiences more clearly in relation to this construct.

## 5. Conclusions

In Australia, 24.9% of recently transitioned veterans will have a diagnosis of PTSD in their lifetime [2]. We know from previous studies that traditional evidence-based treatments are suboptimal for some veterans. Therefore, there is a dire need to supplement these treatments with adjunct interventions. This study exemplifies the power of human–animal relationships and highlights that assistance dogs may be vital for the health and wellbeing of veterans and strongly supports the use of assistance dogs as a treatment adjunct for veterans with PTSD who are open to having an assistance dog. Importantly, veterans themselves reported a life-changing impact of their assistance dog, which adds emphasis to the need to develop health and social environments that support human–animal relationships to enable people to reach their full health potential. It is not clear, given the relatively small number of assistance dogs available and the large numbers of veterans in need of them internationally, how feasible this approach is; however, it is a matter of upmost importance for public health policy. Our findings have implications beyond individual behavioural change and can be used to inform future health promotion, policy, and service delivery in line with the Ottawa Charter [42]. This study suggests that for veterans with PTSD, assistance dogs may be a feasible adjunct intervention.

## Figures and Tables

**Figure 1 ijerph-20-03607-f001:**
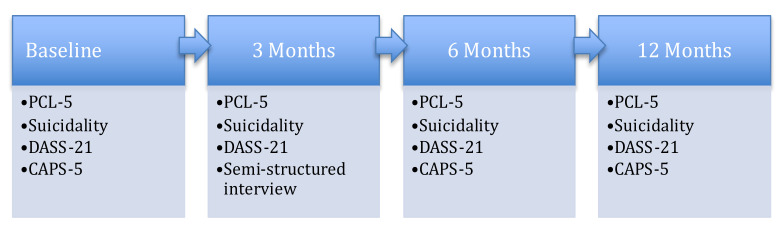
Assessment Timings. Note. PCL-5 = PTSD Checklist. Suicidality = four questions related to suicidal ideation, plans, and behaviours. DASS-21 = Depression Anxiety and Stress Scale—Short Form. CAPS-5 = Clinician-Administered PTSD Scale for DSM-5.

**Figure 2 ijerph-20-03607-f002:**
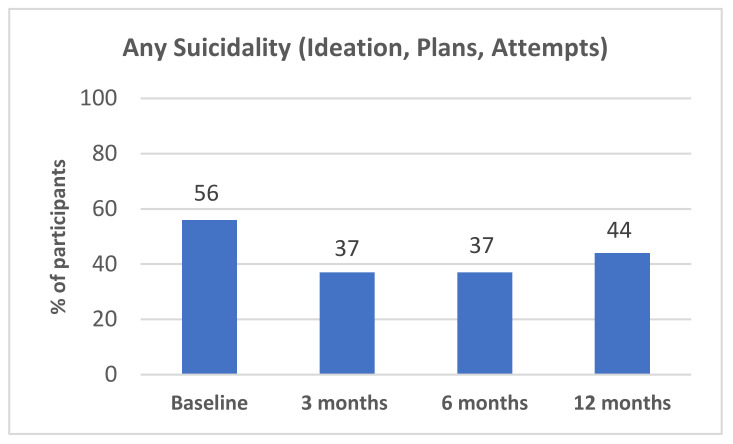
Percentage of Participants Reporting Suicidality (Suicidal Ideation, and/or Plans, and/or Attempts) at Baseline, 3, 6, and 12 Months.

**Table 1 ijerph-20-03607-t001:** Participant Demographic Information.

Variable	Quantitative Data *n* = 16		Qualitative Data *n* = 12	
	*n*	%	*n*	%
Sex				
Male	14	87.5	10	83
Female	2	12.5	2	17
Age				
Range 34–74	*M* = 50.88 (*SD* = 12.88)		*M* = 52.85 (*SD* = 13.3)	
Service				
Navy	2	12.5	2	17
Army	14	75	9	75
Air Force	2	12.5	1	8

**Table 2 ijerph-20-03607-t002:** General Estimating Equation Statistics for Suicidality.

		Exp *β*	95% CI	*p*
Suicidality	Intercept	1.286	[0.48, 3.45]	0.618
	Time			
	Baseline *	(REF)		
	3 months	0.467	[0.21, 1.04]	0.061
	6 months	0.467	[0.16, 1.37]	0.164
	12 months	0.605	[0.18, 2.00]	0.409

* Reference category. CI = confidence interval. *p* = values from Wald tests. Suicidality = suicidal ideation, and/or plans, and/or attempts.

**Table 3 ijerph-20-03607-t003:** Mean and Standard Deviation for Scores on PCL-5 and DASS-21 Depression and Anxiety Subscales.

Measure	Baseline *M* (*SD*)	3 Months *M* (*SD*)	6 Months *M* (*SD*)	12 Months *M* (*SD*)
PCL-5 (/80)	67.69 (16.08)	60.19 (18.86)	54.44 (16.52)	51.56 (18.01)
DASS-21				
Depression (/42)	21.25 (11.84)	16.88 (9.32)	14.13 (10.31)	15.75 (11.59)
Anxiety (/42)	20 (10.30)	12.50 (7.36)	12.25 (8.91)	10.50 (9.14)

Note. PCL-5 = PTSD Checklist; DASS-21 = Depression Anxiety Stress Scale—Short Form.

**Table 4 ijerph-20-03607-t004:** Planned Comparisons for Mean Difference and Effect Size between Baseline and 3, 6, and 12 Months on PCL-5 and DASS-21 Depression and Anxiety Subscales.

Time	Mean Difference	95% CI	Cohen’s *d*
PCL-5 Baseline *			
3 months	7.32 **	[2.85, 11.78]	0.43
6 months	13.07 ***	[8.60, 17.53]	0.81
12 months	15.94 ***	[11.48, 20.40]	0.94
Depression Subscale			
Baseline *			
3 months	4.38**	[0.26, 8.49]	0.41
6 months	7.13**	[3.01, 11.24]	0.64
12 months	5.50**	[1.39, 9.61]	0.47
Anxiety Subscale			
Baseline *			
3 months	7.5***	[3.98, 11.02]	0.84
6 months	7.75***	[4.23, 11.27]	0.8
12 months	9.5***	[5.98, 13.03]	0.98

* Reference category. ** *p* < 0.05. *** *p* < 0.001. CI = confidence interval. PCL-5 = PTSD Checklist.

**Table 5 ijerph-20-03607-t005:** Percentage of Participants Achieving Reliable and Clinically Significant Change in PTSD Symptoms (PCL-5), from Baseline to 3 Months, 6 Months, and 12 Months.

	Baseline to 3 Months		Baseline to 6 Months		Baseline to 12 Months	
	*n*	%	*n*	%	*n*	%
Reliable Change	9	56	13	81	14	88
Clinical Significance	7	44	11	69	10	63

Note. Reliable change threshold = 5 points; clinically significant threshold = 10 points [30]. PCL-5 = PTSD Checklist.

**Table 6 ijerph-20-03607-t006:** General Estimating Equation Statistics for PTSD Case.

		Exp *β*	95% CI	*p*
PTSD Case	Intercept	2.200	[0.72, 6.73]	0.167
	Time			
	Baseline *			
	6 months	0.430	[0.10, 1.84]	0.255
	12 months	0.207	[0.07, 0.66]	0.008 **

* Reference category. *p* = values from Wald tests. ** Significant at *p* < 0.05. CI = confidence interval.

## Data Availability

It is not possible to provide public access to the raw data for this project. This is because of the small size of the state in which this research was undertaken and the small group of highly connected participants, which meant that steps had to be taken to maximise participant anonymity and privacy.

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
