# Peer review of "Effectiveness of Operation K9 Assistance Dogs on Suicidality in Australian Veterans with PTSD: A 12-Month Mixed-Methods Follow-Up Study"

_ijerph, 2023, doi:10.3390/ijerph20043607_

Round 1

Reviewer 1 Report

The article maintains two main intentions, the one that raises the need to promote a substantial advance in the animal-human relationship, based on the certainty of the authors that this benefits vulnerable sectors of society, as is the case of veterans, of the armed forces and their reintegration into society; At the same time that it details and evaluates the experience of its practice, for which it describes in a meticulous way the various resources that were used to achieve it, in the same way, the authors recognize the relative methodological limitations. Its reading provides quality information on a subject that has been little addressed, for which I would recommend its publication.

Author Response

Thank you very much for your feedback, we appreciate you can see the value of in our study. 

Reviewer 2 Report

This is an important topic of high public health relevance. We owe our veterans thoughtful and compassionate services and this study is addresses a promising and relatively practical new direction. The qualitative results are encouraging but the paper should address the commentaries to the O'Haire & Rodriguez (2018) paper that noted directions for future research. This study does not address those concerns but I think still makes a contribution but it would be important to note this and indicate what is needed going forward. Among the concerns noted, for example, is the fact that some veterans found it challenging to manage the needs of their animal and also found that there was undue attention to the fact that they had a service animal. These are issues that can be addressed with ongoing support but should be noted as an important policy need. It should also be noted that given the extremely high rate of PTSD internationally, it is not clear how available this approach would be to the large number of veterans in need. But let's not let the good be the victim of the perfect. This is an important and potentially compassionate approach to address the needs of a vulnerable group in high need of services.

Author Response

Thank you for your feedback. 

We note the paper by O’Haire & Rodriguez (2018) is a quantitative study.  Further, we note the directions for future research in their paper concerning an active comparison control group and have addressed the absence of a control group in our study in the limitations of our study section (line 472).  We have added a sentence to our paper consistent with the commentary of O’Haire & Rodriguez in terms of generalisability only to those veterans who are amenable to receiving an assistance dog (line 480). We have addressed the limitation outlined in their paper regarding biases in self-report by including an objective measure of PTSD severity (CAPS-5), see discussion line 454.

The concern raised regarding addressing previous findings that veterans found it challenging to meet the needs of their animal and undue attention were reported in Yarbrough et al’s (2018) qualitative paper.  We have added in a sentences in our discussion addressing this (line 445) and noting the veterans in our study did not raise similar concerns. We have also added a future research recommendation and policy direction to address this (line 487).  We have also added in a sentence to address the commentary of Yarborough et al (2018) regarding the possible effects observed been due to novelty and O’Haire & Rodrguez (2018) due to attention (line 503).  

We have added a sentence stating it is not clear given the relatively small number of assistance dogs available and the large numbers of veterans in need internationally how available this approach is and is a matter of importance for public health policy (conclusion line (525).

Once again thank you for your feedback and seeing the value in our paper.

Reviewer 3 Report

Paper review:

Effectiveness of Operation K9 Assistance dogs on suicidality in Australian veterans with PTSD:…

General comments:

-          A very interesting, well written, and informative study that, subject to minor changes, I am happy to recommend for publication.

Abstract:

-          Well written, no concerns.

Key words:

-          Keywords: good range of key words, perhaps consider some that allow interested parties to identify ‘veteran’ or similar as this seems to be missing but pertinent.

-          Ottawa? Why was this key word included? It is not mentioned within the paper.

Introduction:

-          I found this to be a very interesting, informative, and well written introduction, with clear research aims.

Methods:

-          Again, I felt this section was well reported. I would have liked to have known more about the Operation K9 programme (dog selection, training, etc) but think the addition of a weblink perhaps to their webpage would suffice.

Results:

-          Figure 2: would prefer the y axis to run from 0 – 100% as looks like suicidality is higher than it actually is if the axis isn’t apprehended. No sure why both the no and yes data is reported. As there are only two categories this adds nothing and makes the figure unnecessarily complicated. I would convert to a % of respondents that reported any indice of suicidality and just show that. Typo in title.

-          Would be useful to report in the methods the cut offs for a small/moderate/large effect size using the Cohen’s D method.

-          Otherwise, I think this is well presented and I have no concerns.

Discussion:

-          Again, well written. I particularly liked the author’s comprehensive appraisal of the limitations.

Tables and figures:

-          Addressed above, otherwise no concerns.

Author Response

Thank you for your feedback and seeing the value in our research, our responses are outlined below in red.

Key words:

-          Keywords: good range of key words, perhaps consider some that allow interested parties to identify ‘veteran’ or similar as this seems to be missing but pertinent. Response: The word veteran has been added as a key word.

-          Ottawa? Why was this key word included? It is not mentioned within the paper.  Response: Ottawa was included in response to the focus of the special issue but as it is only mentioned twice it has been removed from key words.

Methods:

-          Again, I felt this section was well reported. I would have liked to have known more about the Operation K9 programme (dog selection, training, etc) but think the addition of a weblink perhaps to their webpage would suffice.

Response: More information about the programme has been added and link to the website (commencing at line 127).

Results:

-          Figure 2: would prefer the y axis to run from 0 – 100% as looks like suicidality is higher than it actually is if the axis isn’t apprehended. No sure why both the no and yes data is reported. As there are only two categories this adds nothing and makes the figure unnecessarily complicated. I would convert to a % of respondents that reported any indice of suicidality and just show that. Typo in title. Response: These changes have been made.

-          Would be useful to report in the methods the cut offs for a small/moderate/large effect size using the Cohen’s D method. Response: A sentence has been added to methods (line 253). 

Round 2

Reviewer 2 Report

The authors addressed all concerns noted in the prior review which made the paper even stronger. I believe this will be an important contribution to the literature.